# N-Type Printed Organic Source-Gated Transistors with High Intrinsic Gain

**DOI:** 10.3390/nano12244441

**Published:** 2022-12-14

**Authors:** Yudai Hemmi, Yuji Ikeda, Radu A. Sporea, Yasunori Takeda, Shizuo Tokito, Hiroyuki Matsui

**Affiliations:** 1Research Center for Organic Electronics (ROEL), Yamagata University, Jonan 4-3-16, Yonezawa, Yamagata 992-8510, Japan; 2Advanced Technology Institute, School of Computer Science and Electronic Engineering, University of Surrey, Guildford GU2 7XH, UK

**Keywords:** organic transistors, printed electronics, intrinsic gain, amplifiers, contact effects

## Abstract

Source-gated transistors (SGTs) are emerging devices enabling high-gain single-stage amplifiers with low complexity. To date, the p-type printed organic SGT (OSGT) has been developed and showed high gain and low power consumption. However, complementary OSGT circuits remained impossible because of the lack of n-type OSGTs. Here, we show the first n-type OSGTs, which are printed and have a high intrinsic gain over 40. A Schottky source contact is intentionally formed between an n-type organic semiconductor, poly{[N,N′-bis(2-octyldodecyl)naphthalene-1,4,5,8-bis(dicarboximide)-2,6-diyl]-alt-5,5′-(2,2′-bithiophene)} (N2200), and the silver electrode. In addition, a blocking layer at the edge of the source electrode plays an important role to improve the saturation characteristics and increase the intrinsic gain. Such n-type printed OSGTs and complementary circuits based on them are promising for flexible and wearable electronic devices such as for physiological and biochemical health monitoring.

## 1. Introduction

Thin and flexible organic field-effect transistors (OFETs) can be fabricated using low-temperature solution processes, enabling large-area integration at low cost. OFETs fabricated by printing processes can achieve high mobility of over 4 cm^2^/Vs and high printing speeds of 2.4 m/min, depending on the type of printing methods [1]. Furthermore, the appropriate choice of low-toxicity organic solvents such as anisole and 3-methylcyclohexanone may enable devices to be fabricated with an environmentally friendly approach [2,3,4]. A large amount of research in recent years has focused on flexible electronics, including digital circuits [5], temperature sensors [6], humidity sensors [7], pressure sensors [8] and biosensors [9]. The OFETs are important components for amplifying the small signals from the sensors. The conventional OFETs use a gate electric field to modulate the charge density in the active layer between the source and drain electrodes. Ohmic contact has been preferably used to maximize the transconductance. For that reason, an efficient n-type OFET required the source electrode work function to be close enough to the lowest unoccupied molecular orbital (LUMO) of the semiconductor to allow the injection of electrons [10,11]. Suitable charge injection resulted in high current and mobility. It is widely known that if the energy difference is large, a Schottky barrier is formed and the output current of the transistor decreases exponentially with the height of the barrier [12,13,14,15]. Currently, many OFETs demonstrating good charge injection are used in circuit applications. However, the design of multi-stage amplification circuits to obtain high gain increases the circuit complexity and hence increases the production cost and decreases the fabrication yield. In addition, the high power consumption due to high operating voltages has made it difficult to apply them to wearable devices with small energy sources.

The concept of the source-gated transistor (SGT) was first reported by Shannon and Gerstner in 2003 [16]. SGTs structurally resemble FETs but rely on three primary conditions: (1) a source–semiconductor Schottky barrier, (2) the semiconductor layer sandwiched between the source and the insulator/gate, as in the case of a top-gate bottom-contact (TGBC) FET and (3) the gate electrode overlapping with the source electrode to modulate the depletion layer between source and gate. SGTs have three main advantages. The first is the very high intrinsic gain due to the high output resistance [17]; the second is that the current–voltage (I-V) characteristic is independent of channel length, which is a major advantage in inkjet printing where channel length variation is large; the third is the steep sub-threshold slope which reduces the operating voltage and improves transconductance efficiency [18]. In 2013, Rapisarda et al. reported the first organic SGT (OSGT) using p-type organic semiconductors [19]. The operating principle of the OSGT is the same as that of an amorphous silicon SGT [20]. Furthermore, the Nathan research group successfully applied all-printed p-type OSGTs to multi-stage amplification circuits in 2019 [18]. However, no n-type OSGTs have been reported to date. For further improving the gain and power consumption, complementary circuits using p-type and n-type OSGTs are desired. Here, we show n-type printed OSGTs with a high intrinsic gain of over 40. A moderate Schottky barrier was formed by the choice of n-type organic semiconductor, poly{[N,N′-bis(2-octyldodecyl)naphthalene-1,4,5,8-bis(dicarboximide)-2,6-diyl]-alt-5,5′-(2,2′-bithiophene)} (N2200), and bare silver electrode. A blocking layer to mitigate the horizontal electric field at the source edge was introduced to improve the saturation characteristics and the intrinsic gain.

## 2. Materials and Methods

### 2.1. Materials

N-type organic semiconductor polymer, poly{[N,N′-bis(2-octyldodecyl)naphthalene-1,4,5,8-bis(dicarboximide)-2,6-diyl]-alt-5,5′-(2,2′-bithiophene)} (N2200, LUMO: −3.8 eV, M_w_ = 289,730, M_n_ = 150,497), was purchased from Ossila (Sheffield, England) and used without purification [21]. The N2200 was dissolved in o-dichlorobenzene at 0.2 wt%. The organic semiconductor ink was filtered with a 0.44 µm hydrophobic filter after being stirred at 70 °C for 1 day. Cross-linked poly(vinyl phenol) (C-PVP) was prepared by dissolving 10 wt% each of poly(vinyl phenol) (M_w_ = 25,000, Sigma-Aldrich, St. Louis, MO, USA) and poly(melamine-co-formaldehyde) methylated (M_n_ = 432, 84 wt% in 1-butanol, Sigma-Aldrich) in propylene glycol monomethyl ether acetate (PGMEA).

### 2.2. Device Fabrication

Three types of devices denoted as Devices I, II and III were fabricated in the TGBC geometry (Figure 1). The glass substrate was ultrasonically cleaned with acetone, isopropanol, surfactant aqueous solution (Semi-Clean M-LO) and distilled water for 5 min each, and then subjected to UV/O_3_ treatment for 5 min. A C-PVP layer was formed on the glass substrate by spin coating. Parylene (diX-SR, KISCO, Osaka, Japan) was deposited on the C-PVP layer by the chemical vapor deposition (CVD) method to modify the surface roughness and wettability. Source and drain electrodes were fabricated by inkjet printing (DMP-2850, Fujifilm Dimatix, Santa Clara, CA, USA) with a silver nanoparticle ink (NPS-JL, Harima Chemicals Group, Inc., Tokyo, Japan) and sintering at 120 °C for 60 min. Source and drain electrodes of Device I were treated with a self-assembled monolayer (SAM), 4-methylbenzenethiol (4-MBT, Tokyo Chemical Industry Co., Ltd., Tokyo, Japan), to modify the work function. N2200 was deposited to be 50 nm thick by inkjet printing at a stage temperature of 30 °C and a cartridge temperature of 37 °C and annealed at 150 °C for 60 min in a nitrogen glovebox. Parylene with a thickness of 310 nm was deposited as the top dielectric layer, and a silver top gate electrode was printed by inkjet printing.

Device III uses a blocking layer made from C-PVP for mitigating the strong horizontal electric field near the source edge and suppressing the injection of electrons from the source edge to the N2200 (Figure 2). The current density at the edge of the source is known to be influenced by the drain electric field [22,23]. The blocking layer prevents the source from injecting electrons into N2200 under the influence of the source–drain electric field, as the C-PVP is deposited to cover the edge of the source. Prior to printing the blocking layer, a source electrode was printed three times to form a step structure in Device III. C-PVP was deposited by inkjet to cover a part of the source electrode as shown in Figure 1c. The step structure helped to control the spreading of the C-PVP ink. The C-PVP ink could not be blocked by 1–2 layers of the source electrode because of its high wettability. In this study, a Schottky barrier and a horizontal field relaxation structure were used to deliberately suppress the injection of electrons to obtain a high output resistance.

### 2.3. Transistor Characteristics

The characteristics of the organic transistors were measured at room temperature in a glovebox with a 4200-SCS semiconducting parameter analyzer (Keithley Instruments, Solon, OH, USA). The device storage and measurement were performed in a glovebox (nitrogen atmosphere, water < 1 ppm, oxygen < 1 ppm, temperature ranging from 25 to 27 °C). The output resistance, transconductance and intrinsic gain were calculated from the output characteristics as follows.
(1)$$\mathrm{Output}\;\mathrm{resistance}:\;\gamma_{o}=\frac{\partial V_{\mathrm{DS}}}{\partial I_{\mathrm{DS}}}$$
(2)$$\mathrm{Transconductance}:\;g_{m}=\frac{\partial I_{\mathrm{DS}}}{\partial V_{\mathrm{GS}}}$$
(3)$$\mathrm{Intrinsic}\;\mathrm{gain}:\;A_{i}=g_{m}\gamma_{o}$$

## 3. Results and Discussion

Figure 3 shows the transfer characteristics of Devices I, II and III. The characteristics in linear and saturation regimes showed that the current decreases in the order of Devices I, II and III. Device I showed the highest current because of the lowest energy barrier between the 4-MBT modified source electrode and N2200. As for the device structure, Device III showed the lowest current because of the suppression of leakage current by the blocking layer as described below. Since this device uses a Schottky barrier for switching, it is difficult to calculate the mobility. Device I had the highest transconductance of 3.2 µS/cm (Table 1). Device II has a Schottky barrier but shows a high transconductance of 2.7 µS/cm influenced by the high drain electric field. Device III showed the lowest transconductance of 0.64 µS/cm because of the Schottky barrier and the presence of the blocking layer.

The effect of the blocking layer is clearly shown in the output characteristics in Figure 4c. Device III with the blocking layer demonstrates a saturated constant drain current compared to Device I and Device II. Therefore, it is obvious that Device III has a very high output resistance. Table 1 shows the calculated values of output resistance, transconductance and intrinsic gain from Equations (1)–(3), respectively. The intrinsic gain represents the gain of the amplifier where the SGT is combined with an ideal current source. The use of the blocking layer and Schottky barrier increased the output resistance by three orders, while simultaneously decreasing the transconductance by one order. As a result, the intrinsic gain, the product of the output resistance and the transconductance were found to be two orders higher in Device III than Device I. The larger work function tends to increase the output resistance significantly and decrease the transconductance slightly, hence increasing the intrinsic gain. The high drain electric field causes a decrease in the injection barrier from the source to N2200. Therefore, Device II demonstrates the imperfect saturation of drain current above the pinch off despite the high energy barrier. Device III uses a Schottky barrier and blocking layer to prevent the injection of electrons from the edge of the source.

The blocking layer is considered to have two effects [24,25]. (1) The horizontal electric field near the left end of the source/semiconductor interface is mitigated because the electrostatic potential is constant along the surface of the source electrode. Thus, the injection of electrons at the end is suppressed. (2) A large horizontal electric field may exist near the left edge of the source electrode. However, the edge is covered with the blocking layer, which physically inhibits the injection of electrodes there. Therefore, the majority of current flows through a wide area on the top surface of the source electrodes, which results in a constant saturated current.

We note that the barrier height cannot be estimated simply by the LUMO level and the work function because of the vacuum level shift. In other words, the combination of N2200 (LUMO: −3.84 eV) and 4-MBT (work function: −4.0 eV) does not simply mean that the Schottky barrier is absent. The saturation coefficient γ = C_i_/(C_i_ + C_s_) can be computed by considering the insulator and semiconductor specific capacitances, C_i_ and C_s_. Here, γ ≈ 0.15, and there is a large disparity in the change in saturation voltage with gate–source voltage measure in Figure 4c, which returns approximately 0.7 [16]. This indicates that a relatively weak rectifying barrier is present at the source electrode [25].

## 4. Conclusions

In this study, n-type printed organic source-gated transistors (OSGTs) have been demonstrated for the first time. Three types of devices were fabricated by printing processes using two types of electrodes and the blocking layer at the source edge. The blocking layer successfully increased the output resistance by suppressing the effect of the drain electric field and increased the intrinsic gain over 40. The realization of printable high-gain n-type transistors opens up great possibilities for low-power, high-gain, low-cost and simple amplifier circuits for various applications when combined with p-type transistors.

## Figures and Tables

**Figure 1 nanomaterials-12-04441-f001:**
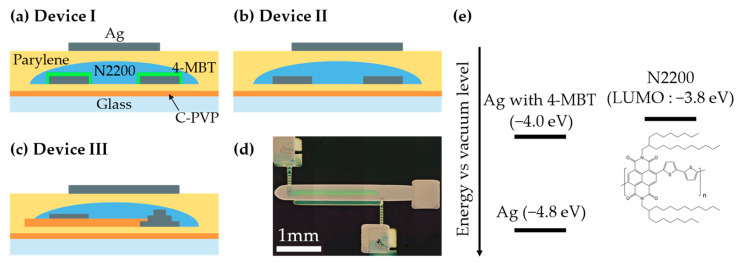
Device structures of (**a**) Device I, (**b**) Device II and (**c**) Device III. (**d**) Optical microscope image of the Device III. (**e**) Molecular structure and energy diagram of N2200 and silver electrodes.

**Figure 2 nanomaterials-12-04441-f002:**
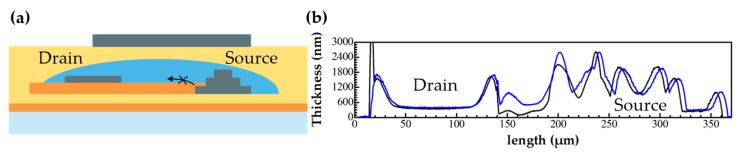
(**a**) Schematics of the role of blocking layer in Device III. (**b**) Cross-sectional plot between source and drain of Device III by Dektak XT. Black line: before printing semiconductor. Blue line: after printing semiconductor.

**Figure 3 nanomaterials-12-04441-f003:**
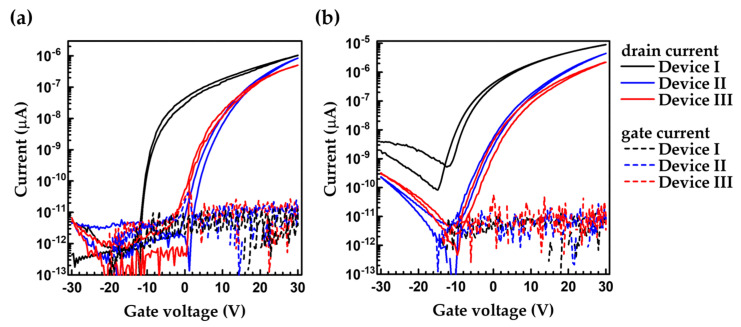
Transfer characteristics at (**a**) *V*_DS_ = 5 V and (**b**) *V*_DS_ = 30V. Solid lines are drain current and dashed lines are gate current.

**Figure 4 nanomaterials-12-04441-f004:**
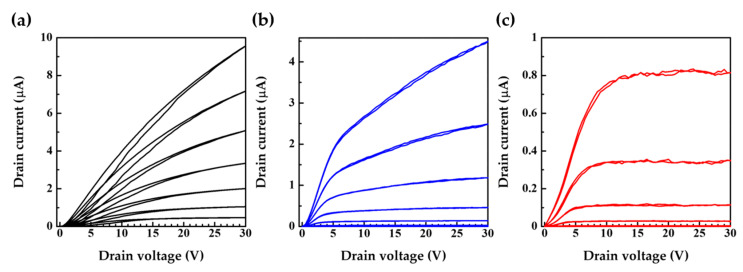
Output characteristics of (**a**) Device I, (**b**) Device II and (**c**) Device III. Gate voltages are 0, 5, 10, 15, 20, 25 and 30 V.

**Table 1 nanomaterials-12-04441-t001:** Output resistance, transconductance and intrinsic gain of the devices. The average and standard deviation were calculated among 4 samples of Device I, 3 samples of Device II and 2 samples of Device III. Only lower limit of the output resistance and intrinsic gain could be estimated in Device III because of the noise and the very small slope in the output characteristics.

	Device I	Device II	Device III
Output resistance (MΩ·cm)	0.69 ± 0.06	2.1 ± 0.1	>100
Transconductance (µS/cm)	3.2 ± 0.5	2.2 ± 0.1	0.4 ± 0.1
Intrinsic gain (-)	2.2 ± 0.4	4.67 ± 0.04	>40

## Data Availability

Not applicable.

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
