# Peer review of "N-Type Printed Organic Source-Gated Transistors with High Intrinsic Gain"

_nanomaterials, 2022, doi:10.3390/nano12244441_

Round 1
Reviewer 1 Report
In their manuscript “N-Type Printed Organic Source-Gated Transistors with High Intrinsic Gain,” Yudai Hemmi and coworkers report on the fabrication of n-channel organic source-gated transistors. In contrast to “regular” field-effect transistors (which typically have source and drain contacts with a small electrical resistance and more or less Ohmic characteristics), source-gated transistors have an intentionally large Schottky barrier at the source, which provides a large intrinsic gain (due to the large output resistance) and makes the current-voltage characteristics independent of the channel length. The latter is an advantage when the transistors are fabricated by methods that cause large variations in the channel length, e.g., by inkjet printing. These benefits come at the expense of a substantially smaller transconductance compared to transistors with Ohmic contacts. There are many reports on n-channel source-gated transistors based on inorganic semiconductors (hydrogenated amorphous silicon, polycrystalline silicon, zinc oxide) and a few reports on p-channel source-gated transistors based on organic semiconductors; this is the first report on n-channel organic source-gated transistors.
The transistors were fabricated in the top-gate staggered device architecture on glass substrates using the semiconducting polymer N2200. For the source and drain contacts, silver nanoparticles were deposited by inkjet printing and sintered at a temperature of 120 degrees Celsius. To alleviate the effect of the lateral electric field, the leading edge of the source contact was covered with an insulating blocking layer (polyvinylphenol). This blocking layer as well as the semiconductor (N2200) and the gate electrodes (silver nanoparticles) were deposited by inkjet printing. Vapor-deposited parylene was used as the gate dielectric.
The transistors have an intrinsic gain of 100, compared to 1.7 in transistors in which the surface of the contacts was modified using 4-methylbenzenethiol (to reduce the Schottky-barrier height) and no blocking layer was used.
The results reported here are certainly interesting and potentially useful, so I recommend publication in Nanomaterials.
A few comments:
In the Abstract and in Section 3, the intrinsic gain is given in units of “V/V”. This makes no sense, since the intrinsic gain is the product of a resistance and a conductance, in units of “(A/V)*(V/A)”. I hope we all agree that the best thing is to give the value without any unit.
In Figure 3, the meaning of the dotted lines is not explained.
The authors mention the “thickness of the electrons”. So how thick is an electron?
What happened to section 4?
According to Table 1, the intrinsic gain is exactly 100, so why do the authors claim in section 5 that an intrinsic gain “over 100” was measured?
“A large amount of research ... has focused” instead of “Many research ... have focused”
“energy difference” instead of “energy gap”
“is the steep subthreshold slope” instead of “is that the steep subthreshold slope”
“in other words” instead of “in other works”
Author Response
We really thank the reviewer for the careful reading, helpful comments and suggestions. We made revisions according to the comments. We provide a point-by-point response as below.
Point 1: In the Abstract and in Section 3, the intrinsic gain is given in units of “V/V”. This makes no sense, since the intrinsic gain is the product of a resistance and a conductance, in units of “(A/V)*(V/A)”. I hope we all agree that the best thing is to give the value without any unit.
Response 1: Thank you for the helpful comment. We agree that the intrinsic gain should be dimensionless. We removed the unit of V/V from our manuscript.
Point 2: In Figure 3, the meaning of the dotted lines is not explained.
Response 2: We noticed that the explanation of the dotted lines were written only in the figure caption and not inside the figure. We added the legend for the dotted lines inside the figure.
Point 3: The authors mention the “thickness of the electrons”. So how thick is an electron?
Response 3: We apologize for the meaningless phrase. We removed the phrase “and thickness of the electrons”.
Point 4: What happened to section 4?
Response 4: We corrected the section number. Now, conclusions section is section 4.
Point 5: According to Table 1, the intrinsic gain is exactly 100, so why do the authors claim in section 5 that an intrinsic gain “over 100” was measured?
Response 5: Related to another reviewer’s comment, we reanalyzed the data for multiple devices. Because of the noise in the output characteristics, it was difficult to estimate the output resistance precisely. What we are sure is the upper limit of the slope of the drain current in the saturation regime. Therefore, we decided to indicate only the lower limit of the output resistance and the intrinsic gain. Now the lower limit of the intrinsic gain is 40.
Point 6: “A large amount of research ... has focused” instead of “Many research ... have focused”
“energy difference” instead of “energy gap”
“is the steep subthreshold slope” instead of “is that the steep subthreshold slope”
“in other words” instead of “in other works”
Response 6: We thank the reviewer for the careful reading. We corrected the expressions accordingly.
Reviewer 2 Report
The study by Hemmi et al. explore the use of n-type N2200 xonjugated polymer in the development of all printed Source-gated transistors (SGTs). They compare three device structures and the corresponding electrical performance. Overall the study is interesting and should be published after minor revisions.
1. Page 1 introduction: The authors discuss the importance of energy barrier and matching electrode WF with LUMO levels of N2200. One simple way to do this is through using electrode interlayers of different metals. This also reduces the contact resistance of the device. the authors should discuss this in the intro and cite studies such as doi.org/10.1063/5.0078907.
2. Figure 1: this figure is hard to follow. The band diagram is very crowded and hard to read. By avoiding the addition of the HOMO level and colouring the levels above the LUMO as well as adding the N2200 structure, hard to follow. I recommend reorganizing or spacing it out.
3. The authors mentioned that devices were characterized in a glovebox. I was wondering if the authors could comment on the air stability of these new device’s structures. It is well known that N2200 is not entirely air stable, does this device structure improve the air stability relative to BGBC or BGTC OTFT configuration?
4. Table 1 should include standard deviations for all the values. How many devices was this data averaged from? Device 3 column should be written in the same form as the rest of the table. (160 instead of 1.6×10^2)
Author Response
We really thank the reviewer for the careful reading, helpful comments and suggestions. We made revisions according to the comments. We provide a point-by-point response as below.
Point 1: Page 1 introduction: The authors discuss the importance of energy barrier and matching electrode WF with LUMO levels of N2200. One simple way to do this is through using electrode interlayers of different metals. This also reduces the contact resistance of the device. the authors should discuss this in the intro and cite studies such as doi.org/10.1063/5.0078907.
Response 1: We thank the reviewer for the helpful suggestion. We added the reference at an appropriate position.
Point 2: Figure 1: this figure is hard to follow. The band diagram is very crowded and hard to read. By avoiding the addition of the HOMO level and colouring the levels above the LUMO as well as adding the N2200 structure, hard to follow. I recommend reorganizing or spacing it out.
Response 2: We thank the reviewer for the helpful comment. We rearranged Figure 1. We hope the new figure is easier to understand.
Point 3: The authors mentioned that devices were characterized in a glovebox. I was wondering if the authors could comment on the air stability of these new device’s structures. It is well known that N2200 is not entirely air stable, does this device structure improve the air stability relative to BGBC or BGTC OTFT configuration?
Response 3: We did not check the air stability of the device carefully. The devices can exhibit the transistor characteristics, while current tend to decrease in air.
Point 4: Table 1 should include standard deviations for all the values. How many devices was this data averaged from? Device 3 column should be written in the same form as the rest of the table. (160 instead of 1.6×10^2)
Response 4: We thank the reviewer for the very important comment. To check the statistics, we reanalyzed multiple devices and calculated the average and standard deviation, while the number of devices was not so large. During that process, we noticed that the error in the estimation of the output resistance and the intrinsic gain in Device III can be significantly large because of the noise in the output characteristics. Since it was difficult to estimate these parameters precisely, we decided to show only the lower limit of these parameters.
Reviewer 3 Report
The manuscript by Hemmi et al. entitled "N-Type Printed Organic Source-Gated Transistors with High Intrinsic Gain" is a high quality article on inkjet printing of organic gate-source transistors. Printable organic electronics is an extremely promising area. This article certainly deserves to be published in Nanomaterials after minor revision.
I want to suggest the following comments and clarify a few concerns which may further improve the article:
1. The following sentence should be clarified (line 148, page 4): “The high drain electric field causes a decrease in the injection barrier and thickness of the electrons from the source to N2200”
2. In my opinion, it should be directly pointed in the text that the blocking layer made from C-PVP
In general, the authors got excellent results and presented a very good manuscript.
Author Response
We really thank the reviewer for the careful reading, helpful comments and suggestions. We made revisions according to the comments. We provide a point-by-point response as below.
Point 1: The following sentence should be clarified (line 148, page 4): “The high drain electric field causes a decrease in the injection barrier and thickness of the electrons from the source to N2200”
Response 1: We are sorry that the sentence did not make sense. The phrase of “and thickness of the electrons” was removed.
Point 2: In my opinion, it should be directly pointed in the text that the blocking layer made from C-PVP
Response 2: We thank the reviewer for the helpful comment. We added an explicit explanation of the material of the blocking layer as “The Device III uses a blocking layer made from C-PVP for mitigating the strong hori-zontal electric field near the source edge and suppressing the injection of electrons from the source edge to the N2200 (Fig. 2).”